# Compact eternal diffractive neural network chip for extreme environments
Yibo Dong [1,3], Dajun Lin [1,2,3], Long Chen[1,2], Baoli Li[1], Xi Chen [1], Qiming Zhang[1], Haitao Luan [1] ✉, Xinyuan Fang [1] ✉ & Min Gu [1] ✉

Artificial intelligence applications in extreme environments place high demands on hardware robustness, power consumption, and speed. Recently, diffractive neural networks have demonstrated superb advantages in high-throughput light-speed reasoning. However, the robustness and lifetime of existing diffractive neural networks cannot be guaranteed, severely limiting their compactness and long-term inference accuracy. Here, we have developed a millimeter-scale and robust bilayer-integrated diffractive neural network chip with virtually unlimited lifetime for optical inference. The two diffractive layers with binary phase modulation were engraved on both sides of a quartz wafer. Optical inference of handwritten digital recognition was demonstrated. The results showed that the chip achieved 82% recognition accuracy for ten types of digits. Moreover, the chip demonstrated high-performance stability at high temperatures. The room-temperature lifetime was estimated to be $1.84 \times 10^{23}$ trillion years. Our chip satisfies the requirements for diffractive neural network hardware with high robustness, making it suitable for use in extreme environments.

Over the years, artificial intelligence (AI) methods have been widely used in recognition[1], autonomous driving[2], scientific research[3–5], human-computer interaction[6], and robotics[7]. With the gradual development of AI capabilities, AI methods are expected to replace manual approaches in extreme environments, such as inclement weather, the deep sea, and space. Neuromorphic hardware is important for the development of AI approaches. For applications in extreme environments, low energy consumption, high robustness, and high speed are important evaluation criteria for neuromorphic hardware. Light is an ideal information carrier because of light-speed wireless passive propagation characteristics, which can help greatly increase the calculation speed while reducing the energy consumption to the level of femtojoules-per-bit[8], and can be used in various calculations, like complex-valued calculations[9], matrix multiplication[10,11] and convolution calculations[12,13]. Compared with electronic parameters (mobility or carrier number), optical parameters (refractive index or transmittance) of materials are generally less sensitive to changes in temperature and humidity. Therefore, photonic components show great potential for use in extreme environments.

Recently, a wave-based optical neural network, the diffractive neural network (DNN)[14], was reported. DNNs have demonstrated superior performance in various AI tasks, such as image recognition[14–20], optical computing[21], phase retrieval[22], adaptive focusing[23], and terahertz pulse shaping[24]. In contrast to waveguide-based optical neural networks[25–28], DNNs mimic the human nervous system in three-dimensional (3D) domains. This feature is realized through the diffraction of waves, thus enabling direct parallel processing of optical image data without converting the data to sequential inputs[29]. This feature enables DNNs to more quickly recognize target objects in extreme environments.

However, because of structural and material limitations, existing DNNs cannot be used in extreme environments. DNNs are composed of cascaded diffractive layers, and currently, 3D printing is widely used to construct DNNs[17,21,22,24,30–33]. However, high robustness and long lifetimes cannot be guaranteed for DNNs made of organic materials. More importantly, the diffractive layers in DNNs are usually spatially separated and operate in the terahertz band[21,24,30–33]. As a result, DNNs are typically on the centimeter scale and cannot be integrated on-chip. However, DNNs integrated on Si wafers have been reported[34–37]. With this method, the three-dimensional networks are designed in two dimensions, thus losing the unique parallel processing advantage for 2D optical images. Therefore, the implementation of a 3D-integrated DNN made with stable materials is highly desirable for applications in extreme environments.

Here, we report an on-chip bilayer DNN for optical inference with virtually unlimited lifetime. Based on double-sided lithography, the two binary phase-modulated diffractive layers in the DNN chip were surface

[1]Institute of Photonic Chips, University of Shanghai for Science and Technology, Shanghai 200093, China. [2]Centre for Artificial-Intelligence Nanophotonics, School of Optical-Electrical and Computer Engineering, University of Shanghai for Science and Technology, Shanghai 200093, China. [3]These authors contributed equally: Yibo Dong, Dajun Lin. ✉ e-mail: haitaoluan@usst.edu.cn; xinyuan.fang@usst.edu.cn; gumin@usst.edu.cn

engraved on both sides of a single-crystal quartz substrate. Therefore, the input wave travels through the quartz during the computing. More than one million neurons were achieved in each layer. Handwritten digit recognition experiments show that the recognition accuracy for ten types of handwritten digits (0~9) is 82%. The consistency and robustness of this chip in fabrication and test sessions are analyzed. The adaptability of this chip to other tasks, including fashion product recognition and phase imaging, has also been verified through simulations. Moreover, the lifetime of the DNN chip was measured. After accelerated aging at high temperatures, the DNN still demonstrates high performance, and the recognition accuracy for two types of digits can be maintained at 100%. The lifetime at room temperature was estimated to be $1.84 \times 10^{35}$ years. This DNN chip strategy satisfies the mass-fabrication requirements for DNN hardware with high robustness and can be used for various AI tasks in extreme environments.

## Results

Figure 1a shows the schematic of the handwritten digit recognition task with the bilayer DNN chip working at a wavelength of 532 nm. The information carried by the neurons was encoded by the phase values and reflected via the pixel heights, which determined the interference of the secondary waves (Fig. 1b). Thus, through the propagation and diffraction of the coherent waves from the input layer to the DNN and finally to the output layer, a feedforward optical neural network was constructed (Fig. 1c). The DNN inference results are displayed by the output layer through the light intensity distribution. Notably, different from the DNNs with separated layers, the signal transmission between the two diffractive layers occurs within the quartz substrate. Therefore, this integration method can ensure the long-term stability of the layer spacing and the diffractive medium, which guarantees the calculation accuracy. Figure 1d shows a digital image of the DNN chip. The diffractive layers are approximately 8.2 mm × 8.2 mm in size.

## TensorFlow-based DNN training

Matrix multiplication is the main mathematical operation used in artificial neural networks. The multiplication of the input and weight matrices of each artificial neural layer reflects the biological signal transmission between neurons through synapses. In DNNs, matrix multiplication operations are implemented optically through the transmission and coherent super-position of the incident coherent waves between the diffractive layers. Therefore, when designing a DNN, a light propagation model between the diffractive layers must be constructed.

Here, we used angular spectrum diffraction to simulate the propagation of the incident light (Supplementary Note 1)[38]. The Fourier transform used in angular spectrum diffraction is suitable for training DNNs with many neurons. Figure 2a illustrates the forward propagation model and error backpropagation model used in the training process. The training was implemented with the TensorFlow 2.0 framework (Google Inc.). We used the Modified National Institute of Standards and Technology (MNIST)[39] handwritten digit database for training. To fabricate the proposed chips, the phase values were trained and binarized (Supplementary Note 2). A training set of 1000 images was used for the DNN training. Based on the principles of DNN, larger training sets can also be used, but accordingly, the training time will increase dramatically. It takes more than 50 h to train a DNN with the same parameters using the full MNIST dataset. More importantly, as shown in Supplementary Fig. 1, the phase distribution resulting from using the full MNIST dataset has a higher frequency compared to a smaller dataset. This will lead to a significant increase in the difficulty of alignment in the optical setup. Therefore, considering the above two factors, we chose a training set of 1000 images.

In addition, the amplitude field of the ten regions was optimized to follow a Gaussian distribution. The area detecting a Gaussian distribution tends to exhibit a more concentrated intensity compared to a typical uniform distribution area, thereby increasing the maximum light intensity

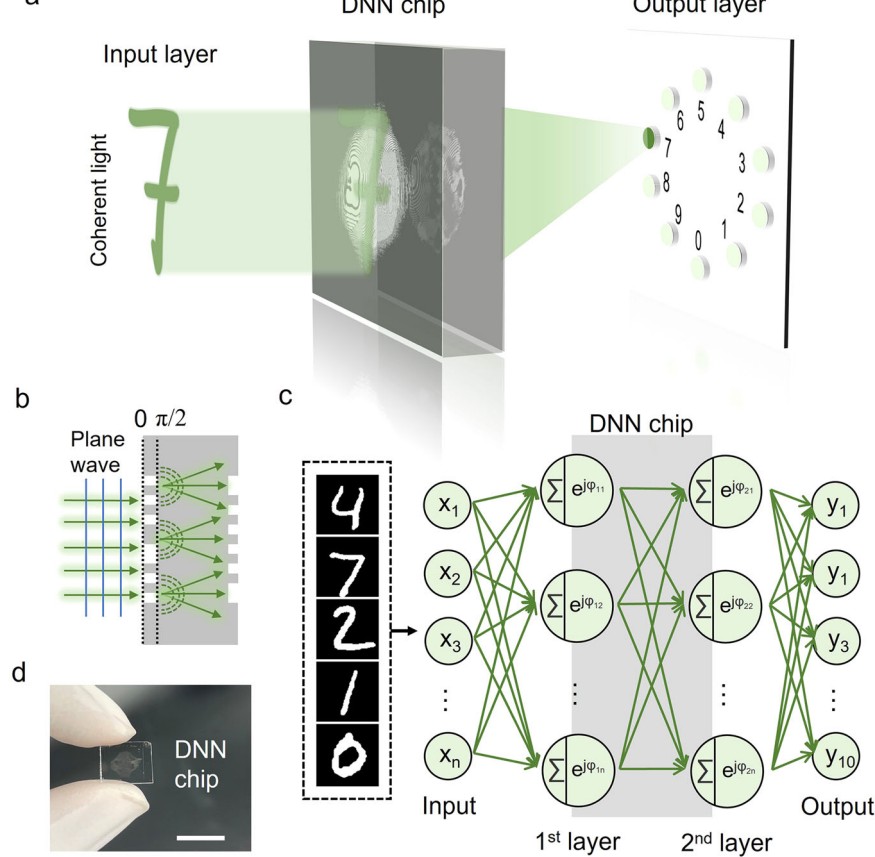

**Fig. 1 | Bilayer diffractive neural network (DNN) integrated on a quartz substrate. a** Schematic of the handwritten digit recognition task with the bilayer DNN. The optical images of the handwritten digits are generated in the input layer. The DNN inference results are displayed by the output layer through the spatial light intensity distribution. The 10 light spots correspond to the digits 0-9. The prediction result is reflected by the light intensity comparison. **b** Schematic of light propagation in the quartz plate. The information carried by the neurons is encoded by the phase values and reflected through the pixel heights. **c** Schematic diagram mathematically describing the physical calculation process of the DNN chip. The optical images of the handwritten digits are generated in the input layer and regarded as the secondary wave source. Before the optical signals are passed to the neurons in the 1st diffractive layer, the coherent superposition of the signals is obtained. Then, the phase values $\varphi$ carried by the neurons are transferred through the light field. The different colors of the neurons represent distinct phase modulations. In our experiment, the diffractive layers were designed with binary phase modulation. Finally, after the light undergoes the same physical process in the 2nd diffractive layer, the output results are generated in the output layer. **d** Digital image of the DNN chip. Each layer contains 1024 × 1024 neurons, and the neuron size is 8 μm × 8 μm. Scale bar, 1 cm.

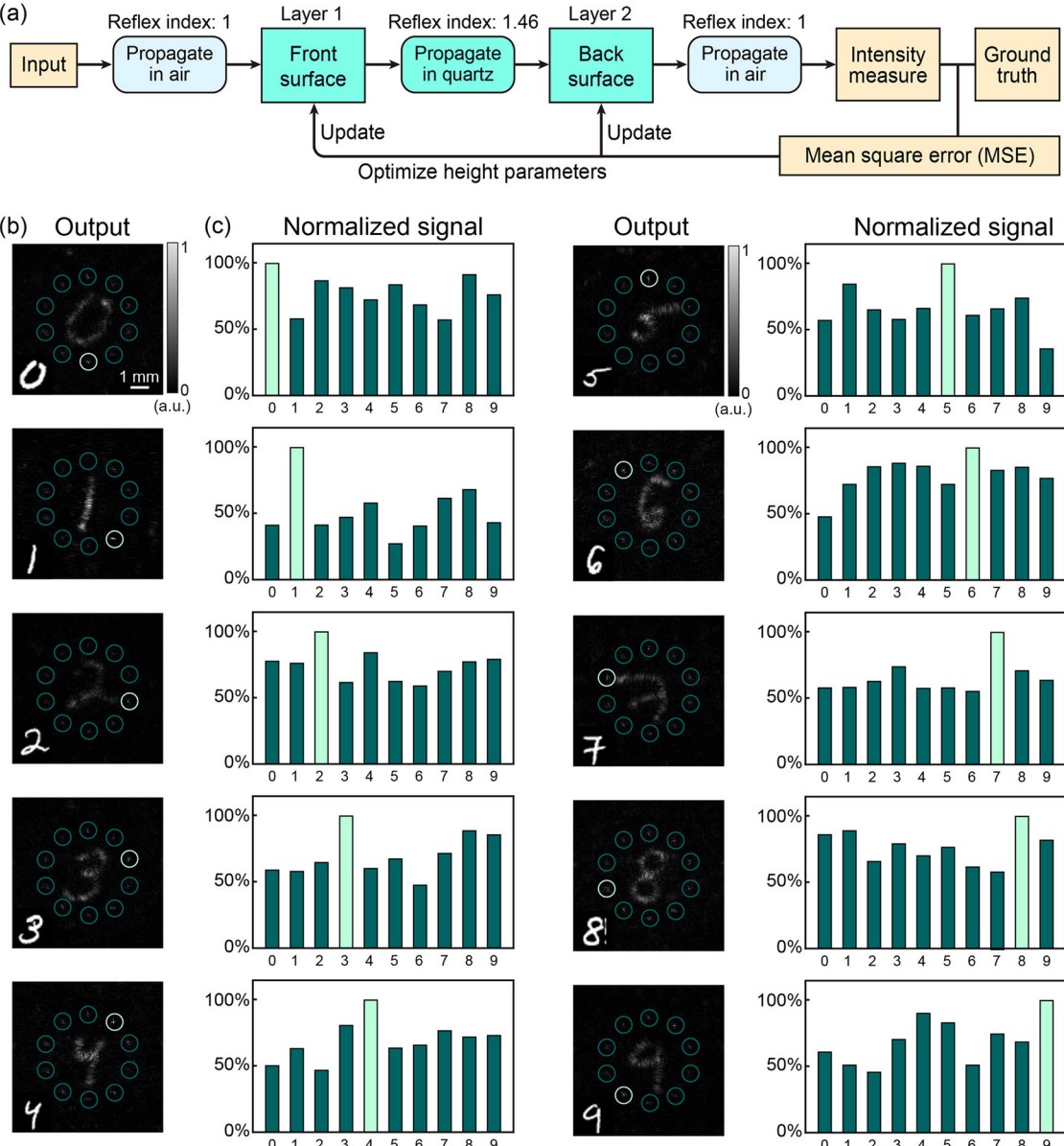

**Fig. 2 | Simulated results of the bilayer diffractive neural network (DNN) chip.**
**a** Training flowchart of the DNN. **b** Light intensity distribution in the output layer.
The light spots in the 10 circled areas represent the handwritten digits from 0 to 9.

The color bar shows the normalized light intensity. **c** Normalized light intensity in
the 10 circled areas shown in (**a**).

density in these regions. Consequently, the camera can capture effective signals more easily, allowing it to operate with shorter exposure time and/or under a lower-laser-power configuration. Thus, the noise in the recorded images can be reduced.

## Analysis of the training results

Supplementary Fig. 2a shows the binary phase distributions obtained in each diffractive layer. Supplementary Fig. 2b illustrates the confusion matrix of the recognition results based on the training set. The results show that the recognition accuracy is 96.1%, with a loss of approximately 0.198. Compared with multilevel phase modulation, we found that binary phase modulation does not appear to affect the accuracy of the DNN. The simulation results presented in Supplementary Table 1 show that the accuracy of the bilayer DNN with binary or 256-level phase modulation differs by only 0.2%.

Figure 2b shows the simulated inference results for 10 typical handwritten digit images. The light spots corresponding to the input digits have the strongest intensity (Fig. 2c), indicating that the DNN successfully

recognized the input digit. In addition, it was observed that the digital images are present in the output layer, indicating that the incident light is not fully modulated by the DNN. This could be due to the binary phase modulation exhibited by the diffractive layers, leading to a decrease in diffraction efficiency. As a result, the zero-order diffraction, which is represented by the digital image, is visible on the output layer. To resolve this issue, the loss function can be modified (refer to Supplementary Fig. 3). However, as constraints increase, the accuracy of DNNs decreases as well, which is contrary to the expected results. Therefore, in our case, the diffraction efficiency and recognition accuracy appear to be two trade-off parameters. Thus, we chose to guarantee a high accuracy and did not change the loss function.

Importantly, we used a commercial single-crystal quartz wafer as the substrate. Because the thickness of the quartz wafer is 500 μm, the layer spacing is fixed at 500 μm. Therefore, the neurons in the DNN are not fully connected. Supplementary Fig. 4 shows that one neuron in the 1st layer is connected to approximately $7 \times 7 = 49$ neurons in the 2nd layer through zero-order diffraction. Although the DNN is not fully connected, the bilayer

DNN still exhibits significantly better performance than the monolayer DNN. We also trained a monolayer DNN, and its recognition accuracy was only 91.2%, with a loss of approximately 0.437 (Supplementary Table 2). The neural network training process can be simply regarded as the optimization of the weight matrices. Therefore, we analyzed the influence of the layer number on the DNN performance based on the degrees of freedom of the weight values (Supplementary Note 3 and Supplementary Fig. 5). The results show that increasing the layer number increases the degrees of freedom of the weight values, thereby increasing the accuracy of the DNN. For a DNN that is not fully connected, improving the neural connections between layers can increase accuracy. This can be achieved by increasing layer spacing or reducing neuron size (Supplementary Fig. 6). The underlying reason can also be explained by the increase in degrees of freedom of the weight values (Supplementary Note 4).

### Performance characterization of the bilayer DNN

We used double-sided photolithography followed by dry etching to engrave the DNN on the surfaces of the quartz plate ("Methods" section). Due to the difference in the refractive indices of quartz and air, pixels with different heights distinctly modulate the phase of the incident light. Supplementary Figs. 7 and 8 show the scanning electron microscopy (SEM) and optical images of the obtained diffractive layer. The diffractive layer pattern is well fabricated on the quartz surface. Plasma dry etching can achieve high-precision engraving with an error of tens of nanometers. Supplementary Fig. 9 shows the height profile of the diffractive layer. The etching depth is approximately 284 nm, which is close to the phase modulation of $\pi/2$ for the 532 nm laser.

Figure 3a shows the optical setup. The laser power was adjusted by a half-wave plate and a polarized beam splitter. We used lenses $L_1$ and $L_2$ to form a 4f system for beam expansion. We adopted a double Fourier transform to generate optical digit images via phase-only spatial light modulation[40]. Then, we filtered out the 0th-order and -1st-order diffraction patterns through spatial filtering, preserving only the 1st-order diffraction pattern for the DNN test. The positions of the input layer, the DNNs chip, and the output layer are shown in Fig. 3b. The dashed orange line following Lens-4 ($L_4$) depicts the conjugate plane of the spatial light modulation. An initial input, represented by an amplitude-only digit pattern '7', propagates across a 5 cm free space and serves as the input for the bilayer DNN chip. Subsequently, after traversing approximately 16.4 cm, the output intensity field with the '7' area brightest is captured by a Complementary Metal Oxide Semiconductor (CMOS) camera. Figure 3c shows the experimental results of the 10 typical handwritten digit images. When a digital image is input into the DNN, the corresponding light spot has the largest intensity (Fig. 3d). We also noted that the unmodulated digit image is obtained, which is consistent with the simulation results. Since the images in Fig. 3c are small, a few enlarged images of the results are shown in Supplementary Fig. 10. Except for the unmodulated digit image, the noise outside the target detection regions is very low, which demonstrates the effects of optimizing the output amplitude field and using many neurons in the design.

Figure 4 shows the confusion matrices of the recognition results based on the test set. Figure 4a shows the simulation results for 1000 images, with the DNN achieving an accuracy of 85.4%. Figure 4b presents the experimental results for 50 images, with the DNN achieving an accuracy of 82%. The accuracy difference between the experimental and simulation results may be due to chip fabrication and measurement errors. The accuracy is not high, mainly because the number of diffractive layers is only two, and the neurons are not fully connected. Simulations show that under the existing bilayer integration configuration of the chip, subsequently reducing the neuron size or increasing the layer spacing can improve the performance (Supplementary Fig. 6). Correspondingly, these optimizations require improvements in fabrication processes. We compare our DNN chip with other reported works in terms of fabrication, integration, robustness, and performance. As shown in Supplementary Table 3, despite some performance differences, our chip shows high robustness and advances in 3D integration compared to other methods.

To verify the chip's capability to perform other AI tasks, we trained the DNN using the Fashion-MNIST dataset[41] with the same chip parameters. The Fashion-MNIST dataset comprises 10 categories of fashion products, which presents greater complexity compared to the MNIST dataset. Supplementary Fig. 11 shows that the DNN chip can achieve approximately 92.2% and 80.1% accuracy for the training set and the test set, respectively. Meanwhile, we also tried a non-recognition AI task with potential practical applications: phase imaging. Phase is invisible, and extracting phase information from light has important applications in wavefront shaping, biological detection, and other fields. In our simulation, the input digital image is phase information (Supplementary Fig. 12a). From the result (Supplementary Figs. 12b, c), we can see that the DNN can directly convert the input phase to intensity information to realize phase imaging. These results indicate that this DNN chip strategy can be used for various AI tasks.

Robustness studies are crucial for chips. We analyze the impact of errors that may occur during fabrication and testing on the performance of DNNs. Alignment of the diffractive layers in DNN is often technically challenging. In our chip, we achieve this through double-sided photolithography, which typically has an overlay accuracy of about 1~2 μm. Based on the simulation results presented in Fig. 5a, b, it can be concluded that alignment errors may cause a decrease in the accuracy of DNN. However, this decrease is slow within the range of 1~2 pixels (8~16 μm). Therefore, it can be inferred that the overlay error of double-sided photolithography has minimal impact on the performance of the DNN chip, and the fabrication process is capable of ensuring high consistency. Then, the thickness of the substrate affects the layer spacing of DNN. We simulated this impact on the accuracy of DNN. As illustrated in Fig. 5c, the accuracy gradually decreases with the thickness deviation, but the rate is slow. Even with a thickness error of ±50 μm (about 10%), the accuracy only decreases by about 2%. Next, we simulated the effects of wavelength shift in the chip's test session. The DNN chip was designed using binary phase modulation of 0 and $\pi/2$, which represents height distributions of 0 and 289 μm for a quartz substrate. Although variations in the incident wavelength only lead to slight changes in phase modulation (Fig. 5d), they can cause a significant decrease in accuracy (Fig. 5e). This is because that the detecting area undergoes magnification or demagnification owing to changes in the numerical aperture. Figure 5f displays intensity patterns at three distinct operating wavelengths, and it is noticeable that the location of the intensity peak shifts. This will lead to a change of the total light intensity in the detection area (white circle area), resulting in recognition errors. However, in the experiment, we can calibrate the position of the output layer to accommodate this change, as well as the positions of the detection areas. In this way, the chip can still demonstrate adaptability to function effectively at other wavelengths. Finally, the test's error also encompasses the misalignment between the input layer and the DNN. As depicted in Fig. 5g, the accuracy of DNNs decreases as the input layer's position shifts. Thus, it is crucial to guarantee a minimal alignment error between the input layer and DNNs. For our chip, the design of DNNs with binary phase modulation helps reduce the difficulty of alignment ("Methods" section).

### Lifetime analysis of the DNN integrated on the quartz substrate

Because of the high melting point and high stability of quartz, the quartz-based optical element has an extremely long and even unlimited lifetime[42]. The reduction in the accuracy of the DNN during accelerated aging was studied. We designed and fabricated several bilayer DNN chips that recognize only two digits (0 and 1), and the accuracy reached 100% (Supplementary Fig. 13). Then, we placed the DNN chip in a box furnace for 2 h at 1400 °C in an air atmosphere. As shown in Fig. 6a, after annealing, the roughness of the surface of the DNN sample increased. The surface roughness of the DNN was $R_a = 0.97$ nm before annealing and $R_a = 20.1$ nm after annealing, thereby increasing the noise in the output layer (Fig. 6b). However, since the prediction results were obtained by comparing the total intensity in the target regions, we found that the increased noise did not affect the recognition results (Fig. 6c), and the DNN accuracy was still 100% for 50 handwritten digit images. After annealing, the mean profile spacing $R_{sm}$ is only hundreds of nanometers, which is much smaller than the

neuron size in the DNN ($8 \times 8 \ \mu m^2$). Therefore, the phase modulation shift in a neuron caused by the increased roughness after annealing is approximately 0. In our experiment, the degradation of the DNN samples at high temperatures is mainly due to the reaction between the quartz and the $Al_2O_3$ boat at high temperatures, resulting in the formation of $Al_2O_3 \cdot SiO_2$[43]. Increasing the annealing time to 3 h completely damaged the samples; the sample shattered and could not be used for handwritten digital recognition.

The estimated lifetime of a device is typically determined based on the Arrhenius equation[42]:

$$\frac{1}{\tau} = k = A \exp(-E_a / k_B T) \qquad (1)$$

where $k$ is the decay rate, $A$ is the frequency factor, $k_B$ is the Boltzmann constant, $E_a$ is the activation energy and $T$ is the absolute temperature. The room-temperature lifetime of a sample can be estimated by extrapolating the lifetime at high temperature to room temperature. We define the time for the DNN sample to be completely damaged as its lifetime. Therefore, as shown in Fig. 6d, the lifetime of the DNN chip at room temperature (300 K) is estimated to be $5.81 \times 10^{42}$ s ($1.84 \times 10^{23}$ trillion years). Even at a temperature of 500 K, the lifetime is still $7.71 \times 10^{23}$ s ($2.44 \times 10^4$ trillion years). The ultralong lifetime of the quartz-based DNN ensures that the chip can operate stably and reliably for a very long time. In addition to high temperatures, we also analyze the impact of other extreme environments on the chip (see Supplementary Note 5). Moreover, unexpected damage, beyond conventional aging, may occur during long-term chip operation.

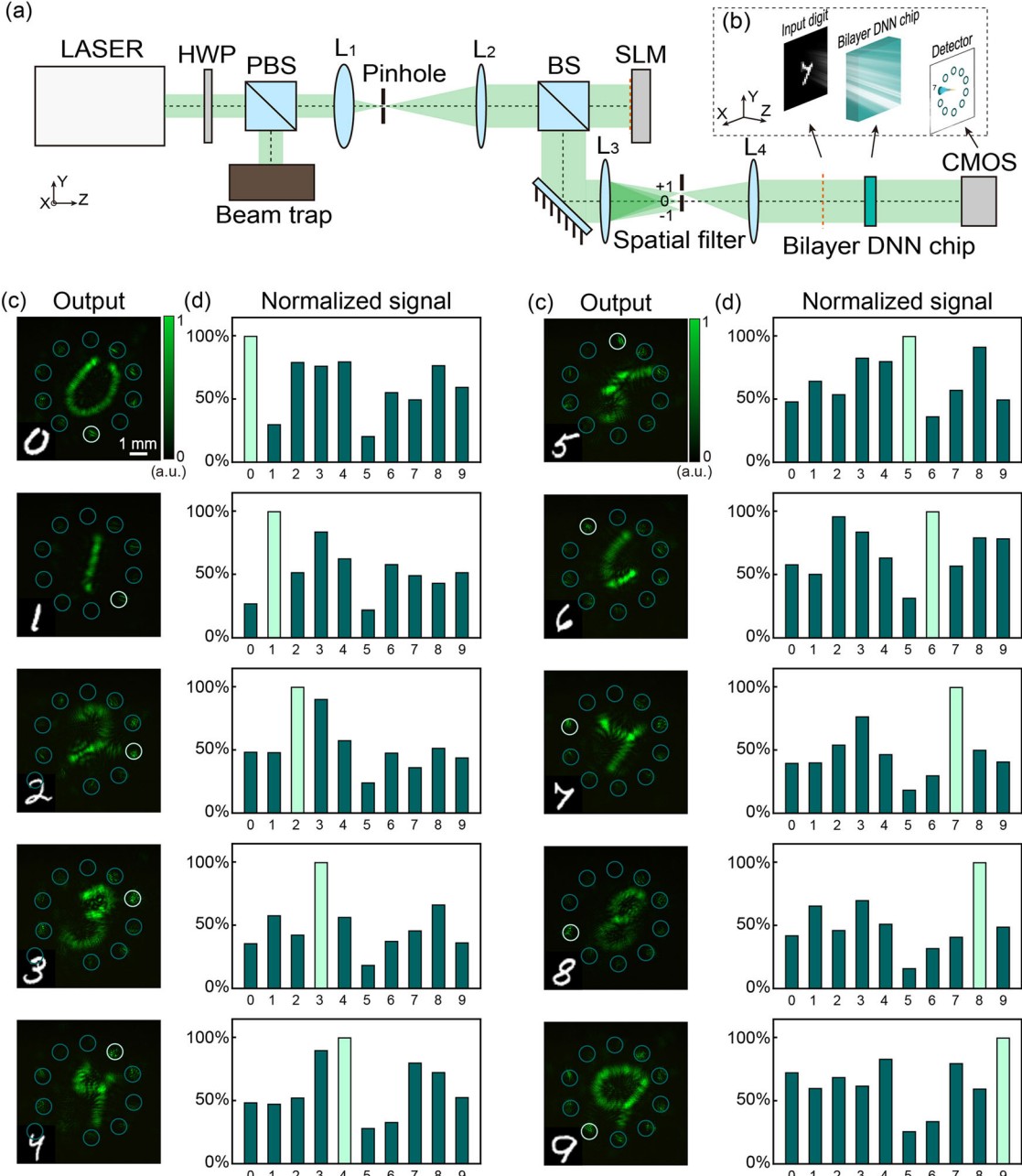

**Fig. 3 | Experimental inference results of the bilayer diffractive neural network (DNN) chip. a** Schematic of the optical setup. We adopted a double Fourier transform to generate optical digit images via phase-only spatial light modulation. PBS: polarized beam splitter. HWP: half-wave plate. BS: beam splitter. SLM spatial light modulation. CMOS complementary metal oxide semiconductor. **b** Schematic of the positions of the input layer, DNNs chip, and output layer in the optical path. **c** Recorded light intensity distribution in the output layer by a CMOS camera. The color bar shows the normalized light intensity. **d** Normalized light intensity in the 10 circled areas shown in (**c**).

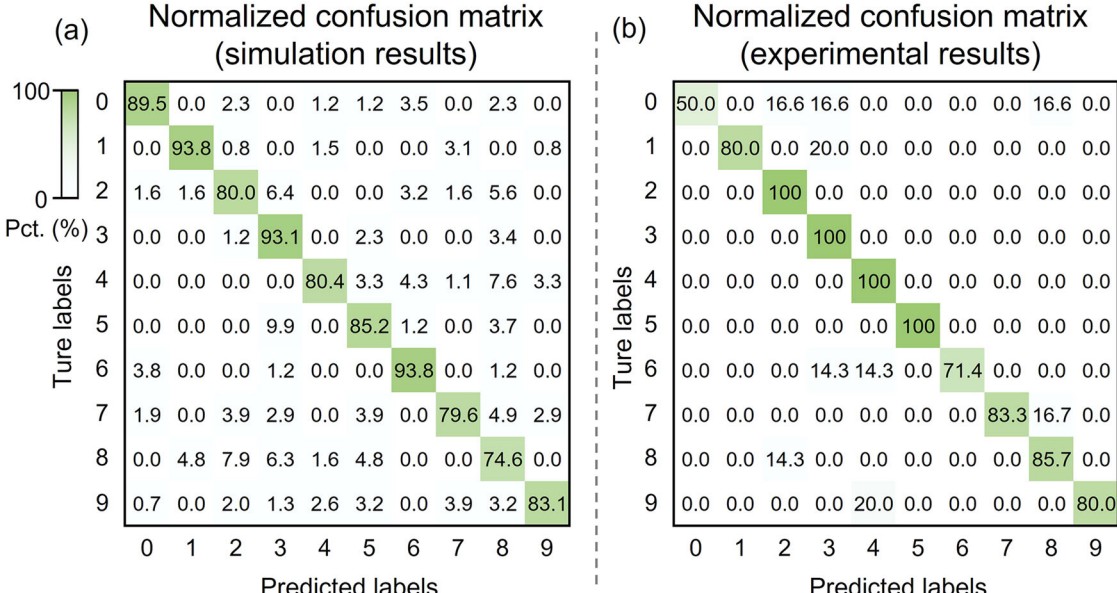

**Fig. 4 | Confusion matrices for the simulation and experimental results based on the test set. Pct. percentage. a** Simulation results. A total of 1000 different handwritten digit images were used. **b** Experimental results. Fifty different handwritten digit images were used.

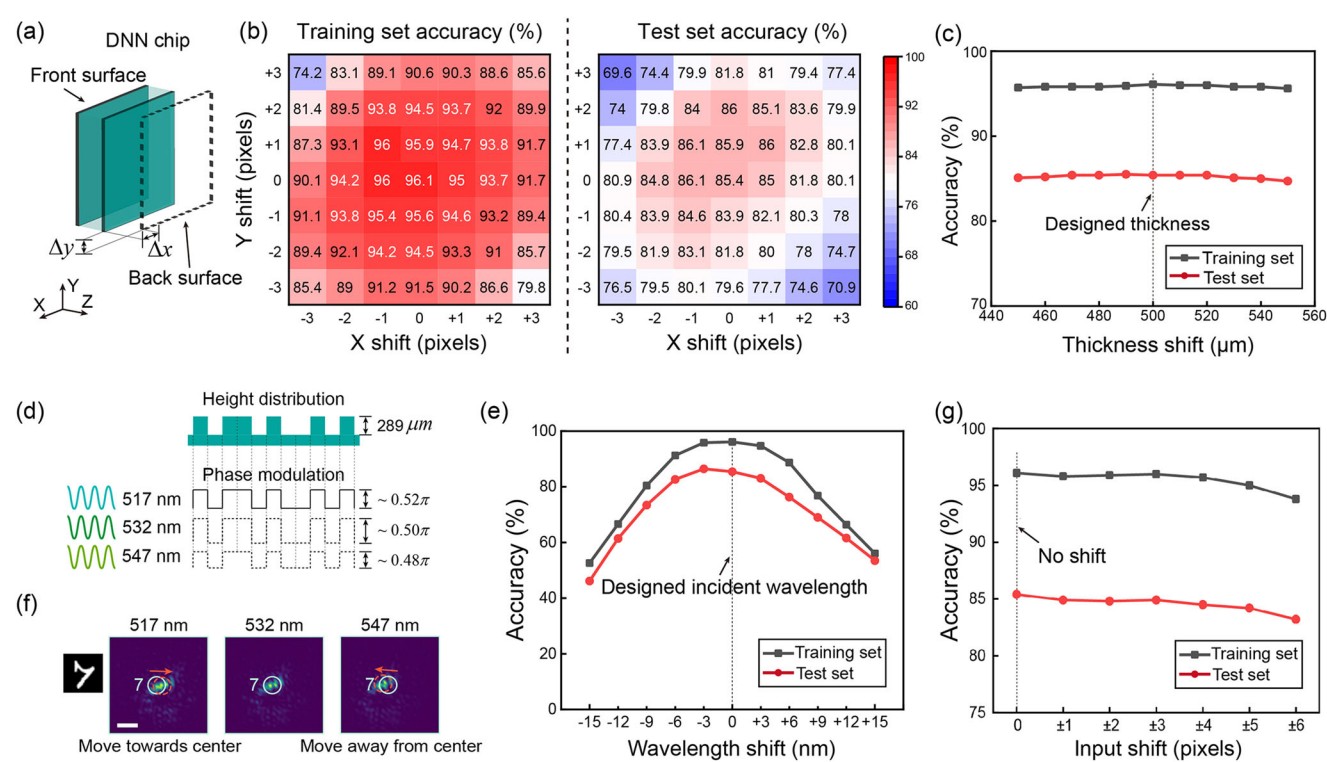

**Fig. 5 | Robustness analysis of the diffractive neural network (DNN) chip in fabrication and test sessions. a** Schematic of the DNN model for the overlay error simulation. $\Delta x$ and $\Delta y$ represent the overlay errors in the x and y directions respectively. **b** The simulated accuracy changes with overlay error for the training set and test set. **c** The simulated accuracy changes with the thickness of the quartz substrate. **d** Phase modulation is different at different working wavelengths. **e** The simulated accuracy changes with working wavelength shift. **f** Magnified output

intensity pattern of the '7' channel at 517 nm, 532 nm, and 547 nm, respectively. Scale bar, 200 μm. The white circle denotes the target detection area. The red dashed circle denotes the actual detection area required for different working wavelengths. The peak moves closer to the center of the output layer with smaller wavelengths (517 nm) and conversely shifts away from the center with larger wavelengths (547 nm). **g** The simulated accuracy changes with the alignment error of the input layer and the DNN chip in the x direction.

We simulated the loss of neuron information resulting from severe damage or wear. Supplementary Fig. 14 shows that the DNN chip can maintain its highest performance when the damaged area is below 20%. This is also a guarantee for the long-term reliable operation of our chip.

## Discussion

In this work, a bilayer DNN chip was integrated on a quartz plate. Our approach based on semiconductor manufacturing technology establishes a more commercial and mature integration solution for DNNs with 3D

**Fig. 6 | Lifetime analysis of the diffractive neural network (DNN) integrated on the quartz substrate. a** Atomic force microscope (AFM) images of the DNN sample before and after annealing. Scale bars, 2 μm. **b** Output images of the DNN chip before and after annealing. The color bar shows the normalized light intensity. **c** Normalized light intensity in the 2 circled areas shown in (**b**). **d** Arrhenius plot of the DNN decay rate. The blue dots represent the experimental results. The orange dots represent the calculated results obtained from the fitting line.

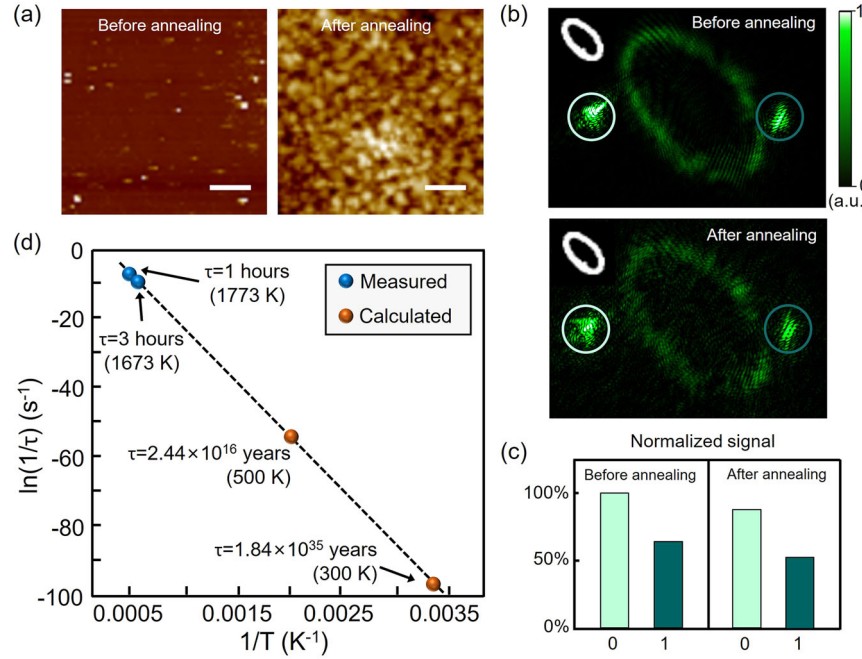

structures. We present an in-depth analysis of the effect of increasing the number of layers and layer spacing on the DNN performance. The robustness of the chip in fabrication and testing is analyzed by simulations. The quartz-based DNN is verified to have an ultralong lifetime and high-performance stability. Thus, it is suitable for long-term operation in various extreme environments, such as strong radiation environments (outer space) and high-pressure environments (deep sea). Other tasks demonstrated based on DNN, including but not limited to beam shaping[24], logical computing[21], and data downscaling[44], could also be theoretically performed using our chip design. This expands the range of potential applications.

We note that non-fully connection constrains the performance of DNN. Therefore, reducing the neuron size or designing wafers with better thickness are important directions for improvement. Additionally, the number of layers in a DNN chip can restrict accuracy improvement. To increase the number of layers, existing bonding techniques can be utilized. Bonding technology allows for the joining of multiple substrates. Laser bonding techniques may be a viable solution for quartz substrates[45,46]. The process can fuse specific areas of quartz, allowing for the selective bonding of two quartz plates. Combined with the double-sided photolithography to realize the alignment of diffractive layers, it is feasible to achieve more diffractive layers. Currently, the lack of nonlinear activation functions between the diffractive layers is a common bottleneck for DNNs, resulting in lower performance compared to deep neural networks. Therefore, it is urgent to solve the problem of inserting nonlinear optical layers between diffractive layers. Our strategy offers a solution by providing the possibility to insert a nonlinear active layer between the diffractive layers. Due to the high stability of quartz, it is possible to deposit nonlinear absorbing materials on its surface. This allows the absorption coefficient of the material to be incorporated into the design of the DNN, resulting in a truly deep DNN. For this purpose, some advanced nonlinear materials, such as two-dimensional materials[47] and perovskite materials[48], can be considered.

Finally, to further achieve the integration of the DNN system, there are pioneering endeavors to learn from. The DNNs chips can be integrated with camera chips[22,49]. In addition, researchers have recently reported the integration of a DNN chip and an electrical neural network chip, demonstrating an analog programmable optoelectronic chip[44]. Besides, we can also consider building a 3D-integrated DNN system based on the vertical-cavity surface-emitting lasers (VCSELs)[29]. VCSELs are micron-sized on-chip light sources that emit light perpendicular to the substrate[50], so VCSEL can be used to generate optical images by constructing a two-dimensional addressable array. The DNNs can be directly integrated on the surface of VCSELs through heterogeneous bonding[51]. The detector array can also be integrated on DNNs with a similar method. In this case, it is technically possible to implement a fully system-integrated DNN chip.

## Methods

### Fabrication of the bilayer DNN

Phase modulation in the diffractive layer is achieved by etching pixels with different depths on the quartz wafer based on the equation $\varphi(\lambda) = 2\pi(n-1)h/\lambda$[52], where $\varphi(\lambda)$ is the target phase modulation of light with a wavelength of $\lambda$, $n$ is the refractive index (1.46) of quartz and $h$ is the etching depth. The fabrication process of the diffractive layer is shown in Supplementary Fig. 15. The etching process is based on the conventional semiconductor manufacturing process. The photolithography equipment is a SUSS MA6 UV lithography machine. Dry etching was carried out with a SENTECH inductively coupled plasma etching system with $SF_6$ gas flow. When fabricating the diffractive layer on the back side of the quartz wafer, double-sided photolithography technology was used to align the two-sided pattern.

### Characterization and annealing of the DNN

The SEM images were acquired with ZEISS Gemini SEM 300 equipment. The height profile of the DNN sample surface was obtained with a Bruker step meter. The AFM images were acquired with a Bruker Dimension Icon microscope. The sample was annealed in an HF-Kejing KSL-1700X box furnace in an air atmosphere.

### Alignment of the input layer and DNNs

In our experiment, we were able to observe the unmodulated optical digital image and the ten light spots generated by the diffraction of DNNs simultaneously on the CCD camera due to binary phase modulation. The digital image indicates the position of the input layer, while the ten light spots indicate the position of the DNNs. Therefore, alignment was achieved by observing their relative positions on the CMOS camera.

## Data availability

The data that support the findings of this study are available on request from the corresponding authors.

## Code availability

The code that supports the findings of this study is available on request from the corresponding authors.

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

## Acknowledgements

We would like to acknowledge support from the National Key Research and Development program of China (2022YFB2804301), the Science and Technology Commission of Shanghai Municipality (Grant No. 21DZ1100500), the Shanghai Municipal Science and Technology Major Project, the Shanghai Frontiers Science Center Program (2021–2025 No. 20), the Shanghai Rising-Star Program (20QA1404100), the Shanghai Sailing Program (23YF1429500) and the National Natural Science Foundation of China (Nos. 11974247, 62005164 and 62005166). This work was also sponsored by the Shuguang Program (23SG41) and Chenguang Program (No. 20CG54) supported by the Shanghai Education Development Foundation and Shanghai Municipal Education Commission.

## Author contributions

Y.B.D and D.J.L contributed equally to this work. Y.B.D. conceived the idea. H.T.L., X.Y.F., and M.G. supervised the project. D.J.L. designed the DNN and organized the experimental results. Y.B.D. fabricated the DNN chip and carried out material characterizations. D.J.L., L.C. Y.B.D, and B.L.L. tested the DNN chip. X.C. and Q.M.Z. helped analyze the results. Y.B.D. and D.J.L. wrote the first draft of the paper. H.T.L., X.Y.F., and M.G. revised the manuscript.

## Competing interests

The authors declare no competing interests.
