## [Peer Review File · Communications Engineering]

Reviewers' comments:

Reviewer #1 (Remarks to the Author):

In the manuscript titled 'A Compact and Robust DNN Chip for Optical Inference in Extreme Conditions,' Dong and Lin et al. introduce a novel design for a deep neural network (DNN) chip. This chip employs a convolutional neural network (CNN) architecture for mask design and is fabricated using a scalable lithography process. The use of a quartz plate as the substrate enhances its robustness and longevity. The authors' experiments, focusing on image recognition tasks with the MNIST dataset, demonstrate the chip's computing capability in optical inference.

The manuscript is coherent and well-articulated. The use of quartz, known for its high chemical stability, is purported to extend the chip's lifetime in extreme environments. This aspect of the design is convincing, suggesting a substantial advancement over existing diffractive neural network implementations, which mostly utilize 3D printing, in terms of durability and reliability. However, while the manuscript demonstrates proof-of-concept experiments and shows some promising results, particularly in terms of long lifetime, I have several concerns regarding performance evaluation and potential applications where robust diffractive optical neural networks are needed. I believe further detail or clarification on these points would enhance the work further:

1. The reported accuracy of 82% using a 1,000 image test set raises questions about the chip's performance on the complete MNIST dataset, which includes 10,000 test images and 60,000 training images. A more comprehensive evaluation, including a comparison with the performance of state-of-the-art digital neural networks on this full dataset, would provide much-needed context and validation for the chip's capabilities. Is there a specific reason for not using the entire dataset? Clarification on this point would be beneficial.
2. The architecture of the digital neural network, as depicted in Figure 3, requires a more detailed description. It is currently unclear whether this architecture directly represents the physical model used, or if it is a one-layer, fully connected network. The test accuracy for the MNIST dataset should easily exceed 90% with a linear model when using the complete training and testing dataset. An explanation of the factors limiting the performance of the digital model is needed. For instance, given the non-fully connected constraint, would increasing the degrees of freedom in the chip help? A more detailed analysis would be helpful to understand the design trade-offs for enhanced performance. Additionally, normalizing the confusion matrix in Figure 3 would facilitate a more direct and meaningful comparison of classification accuracies.
3. One potential concern, which is of course not limited to this work is the potential demanding requirement of the alignment of the optical path for the DNN chip's reasonable performance. However, this aspect is not extensively discussed. The authors should elaborate on the how the system is aligned to guarantee a good performance, the potential challenges and potential limitations of aligning the diffractive layers, particularly considering the stability of the beam path at high temperatures.
4. The chip's performance at high temperatures is impressive. However, expanding on its robustness in

other extreme conditions, such as high pressure environments, would provide a more comprehensive view of its operational range. This should include any limitations or potential issues, particularly concerning the quartz substrate, and how these might be addressed.

5. Further discussions into the aspects in A) consistency of fabrication; 2) robustness to laser wavelength detuning/shift/drift; and 3) performance across different chips of the same design would be insightful. Understanding the extent of fabrication errors, performance variability, and how these factors might affect practical applications of the chip is crucial for assessing its real-world viability.

Besides the major concerns, I also have some minor comments below:

- a. All camera image panels in the figures should include a colorbar for better interpretability.
- b. The phase masks in the supplementary fig. S7 would benefit from colorbars normalized by 2π .

Reviewer #2 (Remarks to the Author):

The paper presents the development of a compact and robust two-layer integrated millimeter-scale diffraction neural network (DNN) chip for optical inference in extreme environments. The use of photons as information carriers in DNNs is a significant advance. It enables reasoning at ultra-low power and at the speed of light, a significant advance in artificial intelligence hardware. The design with two diffraction layers engraved on a quartz wafer improves the reliability of the device. The expected service life at room temperature is extremely long, suggesting its potential for long-term use. The chip's stability at high temperatures and 100% accuracy in recognizing two types of digits are commendable, indicating its suitability for operation in extreme conditions.

While the 82% recognition accuracy across ten digit types is impressive, it represents a somewhat narrow scope of testing. Real-world applications often require processing more complex and diverse data sets.

The article does not sufficiently address the chip's performance with different types of data or its adaptability to other AI tasks beyond digit recognition. This raises questions about its versatility and applicability in various artificial intelligence applications.

The compactness of the device is an advantage, but the article does not address the challenges of scaling this technology to larger networks or integrating it with existing systems and technologies.

To summarize, while the development of this DNN chip represents a remarkable advance in artificial intelligence applications in extreme environments, there are several areas that require further study and clarification.

This article would be better suited for a more specialized journal.

Reviewer #3 (Remarks to the Author):

In this manuscript, the authors present a proposal for a millimeter-scale, compact, and robust dual-layer integrated deep neural network chip with a nearly infinite lifespan. They conducted practical experiments using quartz substrates and demonstrated excellent performance stability even at high temperatures. The innovation is intriguing and well-supported theoretically, and the overall paper is well articulated. The manuscript can be accepted with the following revisions:

-In Figs. 2 and 3, despite achieving a high level of training accuracy, there appears to be some crosstalk in various receiver channels. Could the authors provide a brief analysis and propose corresponding improvement strategies?

-The authors claim that "In addition, to reduce the noise in the output layer, the amplitude field in each region was optimized to a Gaussian distribution to increase the maximum light intensity." Could the authors further elaborate on the principle behind this approach?

-Quartz substrates are employed for diffraction propagation in this manuscript. Could the authors briefly analyze how mechanical errors in the thickness of the quartz substrate affect the experimental results?

-The paper's title suggests permanence in extreme environments. In such scenarios, it is essential to analyze unexpected incidents beyond temperature indices, such as the impact of mechanical damage to the device. Please provide a brief robust analysis of these aspects.

-I suggest that the authors cite some representative papers on metamaterials designed by machine learning and deep learning, including [Nat Commun 13, 2694 (2022)], [Light Sci Appl 12, 82 (2023)], [Prog Electromagn Res 175, 139-147, (2022)], [Prog Electromagn Res 175, 81-89, (2022)]. This will help contextualize their work within the existing literature and provide additional support for the novelty and significance of their contributions.

Manuscript ID: COMMSENG-23-0475A

Title: Compact eternal diffractive neural network chip for extreme environments

Dear Reviewers,

Thank you very much for reviewing and commenting on our manuscript. Addressing your comments and concerns has helped us to improve the quality of the manuscript and increase its appeal to an audience with different backgrounds. Please find below a point-by-point response to your comments. The revised parts have been marked in red in the Main Document.

Reviewer #1

In the manuscript titled 'A Compact and Robust DNN Chip for Optical Inference in Extreme Conditions,' Dong and Lin et al. introduce a novel design for a deep neural network (DNN) chip. This chip employs a convolutional neural network (CNN) architecture for mask design and is fabricated using a scalable lithography process. The use of a quartz plate as the substrate enhances its robustness and longevity. The authors' experiments, focusing on image recognition tasks with the MNIST dataset, demonstrate the chip's computing capability in optical inference.

The manuscript is coherent and well-articulated. The use of quartz, known for its high chemical stability, is purported to extend the chip's lifetime in extreme environments. This aspect of the design is convincing, suggesting a substantial advancement over existing diffractive neural network implementations, which mostly utilize 3D printing, in terms of durability and reliability. However, while the manuscript demonstrates proof-of-concept experiments and shows some promising results, particularly in terms of long lifetime, I have several concerns regarding performance evaluation and potential applications where robust diffractive optical neural networks are needed. I believe further detail or clarification on these points would enhance the work further:

Response: Thank you for your time and attention to our manuscript. We greatly appreciate your positive comments which encourage us so much. Your detailed and professional critiques and advices are very helpful to us.

Comment 1: The reported accuracy of 82% using a 1,000-image test set raises questions about the chip's performance on the complete MNIST dataset, which includes 10,000 test images and 60,000 training images. A more comprehensive evaluation, including a comparison with the performance of state-of-the-art digital neural networks on this full dataset, would provide much-needed context and validation for the chip's capabilities. Is there a specific reason for not using the entire dataset? Clarification on this point would be beneficial.

Response: Thank you for raising this concern. Larger training sets can also be used to train the DNNs with longer training time. We used a small training set mainly for the following reasons. The number of neurons we used in each layer exceeds 1 million (1024×1024 pixels), which can bring a high spatial bandwidth product, thus effectively improving the output representation effect. However, the large number of neurons also causes a significant increase in the amount of training computation, resulting in a training time of over 50 hours. More importantly, as shown in Figure R1, the phase distribution resulting from using the full MNIST dataset has a higher frequency compared to a smaller dataset. The higher frequency patterns in the DNN chip will lead to a significant increase in the difficulty of alignment. Therefore, considering the above two factors, we chose a training set of 1000 images.

Meanwhile, as you suggested, we compared the performance and feature differences of our work with other works that demonstrated advances on the fabrication or integration of DNNs (Table R1). The common features of all works are that they all fabricated DNN samples in the experiment and the DNNs operate in the visible or near-infrared band. Compared to Ref. [2] with the same task, the accuracy of our DNN is slightly lower. This may be due to the small number of training images and the less diffractive layers. In terms of fabrication and integration of DNNs, our chip demonstrates high robustness and advances in 3D integration compared to other methods.

Figure R1. Frequency study of the phase distribution of two DNNs trained with datasets of 1000 images and 55000 images, respectively. **a** Schematic illustration of the picked regions of the two DNNs. **b** Magnified phase distribution of the regions shown in **a**. Region size: 80×80 pixels. **c** Phase distribution along the horizontal lines shown in **b**.

Table R1. Comparison of the reported DNNs with this work

Ref.	Implementation of DNN	Diffractive layers	Number of layers	Robustness of DNN	Tasks	Test accuracy
1	Silicon metasurface	Separated	2	Moderate	Digital recognition (6 types)	90%
2	Nanofabrication on quartz plates	Separated	5	High	Digital recognition (10 types)	84%
3	3D laser printing in lithium niobate	/	1	Moderate	Odd and even classification (2 types)	90%
4	TiO ₂ metasurface	/	1	Moderate	Digital recognition (4)	93.75%

					types)	
5	Si photonics	Integrated (2D integration)	3	Moderate	Iris plants classification (4 types)	90%
6	3D organic laser printing	Integrated (3D integration)	4	Low	Direct retrieval	/
This work	Double-sided nanofabricati on on a quartz wafer	Integrated (3D integration)	2	High	Digital recognition (10 types)	82%

References

- [1] He C, et al. Pluggable multitask diffractive neural networks based on cascaded metasurfaces. *Opto-Electronic Advances* 7, 230005 (2024).
- [2] Chen H, et al. Diffractive Deep Neural Networks at Visible Wavelengths. *Engineering* 7, 1483-1491 (2021).
- [3] Chen P, et al. Laser nanoprinting of 3D nonlinear holograms beyond 25000 pixels-per-inch for inter-wavelength-band information processing. *Nature Communications* 14, 5523 (2023).
- [4] Luo X, et al. Metasurface-enabled on-chip multiplexed diffractive neural networks in the visible. *Light: Science & Applications* 11, 158 (2022).
- [5] Fu T, et al. Photonic machine learning with on-chip diffractive optics. *Nature Communications* 14, 70 (2023).
- [6] Goi E, Schoenhardt S, Gu M. Direct retrieval of Zernike-based pupil functions using integrated diffractive deep neural networks. *Nature Communications* 13, 7531 (2022).

Revised content:

Lines 129-137, Page 6, Main Document

Lines 243-246, Pages 11-12, Main Document

Figure S1, Page 11, Supporting Information

Table S3, Page 29, Supporting Information

Comment 2: The architecture of the digital neural network, as depicted in Figure 3, requires a more detailed description. It is currently unclear whether this architecture directly represents the physical model used, or if it is a one-layer, fully connected network. The test accuracy for the MNIST dataset should easily exceed 90% with a linear model when using the complete training and testing dataset. An explanation of the factors limiting the performance of the digital model is needed. For instance, given the non-fully connected constraint, would increasing the degrees of freedom in the chip help? A more detailed analysis would be helpful to understand the design trade-offs for enhanced performance. Additionally, normalizing the confusion matrix in Figure 3 would facilitate a more direct and meaningful comparison of classification accuracies.

Response: Thank you for your professional question. The physical model of our chip is a non-fully connected bilayer DNNs because the substrate used is a commercial single crystal quartz wafer which has a small thickness of 500 μm . We have modified Figure 3a in the Main Document and described it in more detail. As shown in Figure R2 below, we mark the positions of the input layer, the DNNs chip, and the output layer in it. The dashed orange line after L4 represents the conjugate plane of the SLM. An initial input, represented by an amplitude-only digit pattern '7', is propagated over a 5 cm free space. This pattern serves as the input to the bilayer DNN chip. Then, after traveling approximately 16.4 cm, the output intensity field with the brightest '7' region is captured by a CMOS camera.

Figure R2. Schematic of the optical setup.

As you suggested, we examined the impact of degrees of freedom, such as layer spacing and pixel size (i.e. neuron size), on the accuracy of DNNs under the non-fully connected constraint. Figures R3a and b demonstrate that accuracy improves as pixel size decreases. This is described by the equation, $\sin\theta = m \frac{\lambda}{d}$, where, m donates the order of the diffraction, λ represents the wavelength of the incident light, d refers to the spacing between the diffracting elements in the grating. A larger diffraction angle

contributes to more connections between the two layers, thereby leading to higher training accuracy. Meanwhile, the layer spacing distance within the DNN chip, namely the gap between the double-side patterns, also plays a significant role. Figures R3 c and d illustrate that a larger layer spacing distance leads to better accuracy. This improvement is attributed to the wider gap improves the neuronal connection.

Figure R3. Simulation of the impact of neuron size change and layer spacing change on the performance of the DNNs chip. A training set and a test set with 1000 images are used. **a** Illustration depicting DNNs with different pixel sizes while maintaining an identical chip size. **b** Comparison of training accuracy against varying pixel sizes. **c** Illustration of layer spacing change for DNN. **d** Accuracy changes under different layer spacing.

At last, we have revised the confusion matrixes to normalized display. The revised figures are shown below.

Figure R4 (Figure 4 in the Main Document). Confusion matrices for the simulation and experimental results based on the test set. Pct., percentage. a Simulation results. A total of 1000 different handwritten digit images were used. b Experimental results. Fifty different handwritten digit images were used.

Revised content:

Lines 189-191, Page 9, Main Document

Figures 3a and b, Page 10, Main Document

Lines 218-224, 239-241, Page 11, Main Document

Figure 4, Page 12, Main Document

Figure S6, Page 17, Supporting Information

Comment 3: One potential concern, which is of course not limited to this work is the potential demanding requirement of the alignment of the optical path for the DNN chip's reasonable performance. However, this aspect is not extensively discussed. The authors should elaborate on the how the system is aligned to guarantee a good performance, the potential challenges and potential limitations of aligning the diffractive layers, particularly considering the stability of the beam path at high temperatures.

Response: Thank you for your insightful comment. Alignment is a crucial factor that affects the performance of DNNs. Our experiment considers two aspects of alignment: the alignment of diffractive layers and the alignment of the input layer and the DNNs.

In the revised manuscript, we analyze the impact of these two types of alignment errors on the performance of DNNs and provide a detailed explanation of how we achieved alignment in the experiment.

First, for the alignment of the diffractive layers, this can be realized easily in our work, because double-sided photolithography can achieve the alignment of double-sided patterns through the designed registration marks. The overlay accuracy typically reaches about 1~2 μm , which is smaller than the neuron size (8 μm ×8 μm) we used. We also simulated the impact of the overlay error between the two layers on the accuracy. As shown in Figure R5, within an error of 1~2 pixels (8-16 μm), there is no significant impact on the accuracy.

Figure R5. Simulation of the impact of photolithography overlay error on the performance of DNN. **a** Schematic of the DNNs model. Δx and Δy represent the overlay errors in x and y directions respectively. **b** The accuracy of DNNs changes with overlay error for the training set and test set.

Then, we conducted simulations to evaluate the impact of misalignment of the input layer and the diffractive layers on the accuracy (Figure R6). It can be seen that as the position of the input layer shifts, the accuracy of DNNs also decreases. Therefore, it is crucial to ensure a small alignment error between the input layer and DNNs. In our work, the design of DNNs with binary phase modulation helps reduce the difficulty of alignment. Due to binary phase modulation, the unmodulated optical handwritten digital image and the ten light spots generated by the diffraction of DNNs can be observed simultaneously on the CMOS camera. The digital image represents the position of the input layer, while the ten light spots represent the position of the DNNs. Therefore, we achieved the alignment by directly observing their relative positions on

the camera.

Figure R6. Simulation of the impact of alignment error between the input layer and DNNs on the performance of DNN. a Schematic of the simulation model. Δx and Δy represent the alignment errors in x and y directions respectively. **b** The accuracy of training set and test set changes with the alignment error.

At high temperatures, for our chip, the alignment of the diffractive layer is not affected because the two layers are integrated directly on a quartz wafer. However, high temperatures can cause changes in the density of the light propagation medium and the thermal expansion of the quartz. Therefore, if the DNN chip is operated directly in a high temperature environment, the high temperature may affect the optical path between the input layer and the DNN, as well as the optical path between the diffractive layers. For this reason, we may need to take these changes into account when training DNNs.

Finally, regarding potential challenges, in the future, as DNNs systems shrink and integrate, the difficulty of aligning their optical paths will continue to increase. Manual alignment will not be feasible, and an automated alignment calibration system will need to be developed. Further, once the system of DNNs is fully integrated on-chip, the alignment of the optical path will no longer be a mechanical problem, but will become a micro-nano-processing problem, which can be solved through photolithography. The first is the double-sided photolithography used in our work. Further, by introducing advanced bonding techniques, such as laser bonding[1], the integration and alignment of more diffractive layers will also be possible. This method is compatible with

conventional CMOS manufacturing and can achieve large-scale fabrication. Another feasible route is laser 3D printing. Highly stable materials with nanometer resolution, such as optical-grade glass, can now be 3D printed using two-photon polymerization [2]. For 3D printing, it can naturally achieve the alignment [3]. Then, for the alignment of the input layer and DNNs, once the chip-scale light source is used, the alignment of the input layer and DNNs can also be achieved through photolithography. For instance, the addressable vertical-cavity surface-emitting laser (VCSEL) array is an ideal light source for DNNs [4]. DNNs can be integrated directly onto the VCSEL surface using double-sided photolithography and laser bonding, eliminating the need for optical path alignment.

References

- [1] Huang H, Yang L-M, Liu J. Ultrashort pulsed fiber laser welding and sealing of transparent materials. *Appl Opt* 51, 2979-2986 (2012).
- [2] Bauer J, Crook C, Baldacchini T. A sinterless, low-temperature route to 3D print nanoscale optical-grade glass. *Science* 380, 960-966 (2023).
- [3] Goi E, Schoenhardt S, Gu M. Direct retrieval of Zernike-based pupil functions using integrated diffractive deep neural networks. *Nature Communications* 13, 7531 (2022).
- [4] Gu M, Dong Y, Yu H, Luan H, Zhang Q. Perspective on 3D vertically-integrated photonic neural networks based on VCSEL arrays. *Nanophotonics* 12, 827-832 (2023).

Revised content:

Lines 267-276, Page 13, Main Document

Lines 292-297, Pages 13-14, Main Document

Figures 5a, b and g, Page 14, Main Document

Conclusion, Page 17, Main Document

Methods, Page 19, Main Document

Comment 4: The chip's performance at high temperatures is impressive. However, expanding on its robustness in other extreme conditions, such as high pressure environments, would provide a more comprehensive view of its operational range. This

should include any limitations or potential issues, particularly concerning the quartz substrate, and how these might be addressed.

Response: We greatly appreciate your professional comment. We analyzed the chip's robustness in other extreme environments. We have analyzed the robustness of the chip in other extreme environments, including high-pressure environments, strong radiation environments, and high-humidity environments. We have added the following analysis to the supporting information.

Firstly, the high-pressure environment is analyzed. The DNN chip was fabricated by surface carving on a single-crystal quartz wafer using plasma dry etching. This process does not alter the lattice structure of quartz, so we think that the chip's robustness under high pressure can be compared to that of a silica crystal without structures. Typically, the threshold for high pressure to alter the lattice of a silica crystal is above 25 GPa [1]. Changes in the crystal lattice will lead to changes in the refractive index, which can be regarded as the failure of the chip. Therefore, it is speculated that this chip can withstand high pressures above GPa. Additionally, in strong radiation environments, defects may form in quartz [2], which could scatter light. This scenario is similar to the experimental results obtained after high-temperature annealing (refer to Figure 6b and c in the Main Document). After annealing, the surface of the chip becomes rough, which scatters incident light as well. However, the inference accuracy of the chip is not affected. This is because the inference results are obtained by comparing the relative strength of the ten light spots. Therefore, we think that the radiation-induced defects in quartz will also not affect the chip's performance under certain radiation limits. Third, for high humidity environments, due to the high stability of silica, we think that the humidity will not affect the chip's performance.

[1] Binggeli N, Chelikowsky JR. Structural transformation of quartz at high pressures. *Nature* 353, 344-346 (1991).

[2] Wang B, Yu Y, Pignatelli I, Sant G, Bauchy M. Nature of radiation-induced defects in quartz. *The Journal of Chemical Physics* 143, 024505 (2015).

Revised content:

Supplementary Note 5, Page 10, Supporting Information

Comment 5: Further discussions into the aspects in A) consistency of fabrication; 2)

robustness to laser wavelength detuning/shift/drift; and 3) performance across different chips of the same design would be insightful. Understanding the extent of fabrication errors, performance variability, and how these factors might affect practical applications of the chip is crucial for assessing its real-world viability.

Response: Thank you for your comment. (1) Consistency of fabrication. The consistency of fabrication is primarily affected by the accuracy of double-sided photolithography, which can typically achieve an overlay accuracy of about 1-2 μm . In our response to Comment 3, we simulated the impact of overlay errors (i.e. alignment errors) on the chip's performance and found that they have little effect. Therefore, the fabrication method can ensure high consistency. (2) Robustness to laser wavelength detuning/shift/drift. The bilayer DNN is designed using binary phase modulation of 0 and $\pi/2$, representing height distributions of 0 and 289 μm for quartz substrate, respectively. Although variation in incident wavelength only leads to slight changes in phase modulation (Figure R7a), this can cause a significant decrease in accuracy (Figure R7b). This is because that the detecting area undergoes magnification or demagnification owing to changes in the numerical aperture (NA). As wavelengths vary, each with its unique wave number, they induce alterations in the intensity pattern on the output layer. Figure R7c shows intensity patterns at three distinct operating wavelengths. It's noticeable that the location of the intensity peak shifts: it moves closer to the center of the output layer with smaller wavelengths (517 nm) and conversely shifts away from the center with larger wavelengths (547 nm). The movement of the intensity peak leads to the change of the total light intensity in the white circle area, resulting in a rapid decline in accuracy. However, in experiment, we can calibrate the position of the output layer, namely the position of the CMOS camera, to accommodate this change, as well as adjusting the detecting area for each channel. In this way, the chip can still demonstrate adaptability to function effectively at other wavelengths. (3) Performance across different chips of the same design. We trained a DNN with Fashion-MNIST datasets under the same design. This dataset comprises 10 categories of fashion products and is more challenging than handwritten digits. The DNN chip achieved an accuracy of approximately 92.2% and 80.1% for the training and test sets, respectively, which is lower than that of MNIST handwriting digit recognition (Figure R8). This demonstrates that our approach for integrated DNNs chips can be applied to various AI tasks, although the accuracy may vary depending on the task's complexity. In addition,

because of the same physical diffractive principles, other tasks demonstrated based on DNN, like phase imaging[1], beam shaping[2], optical computing[3] and data downscaling[4], could also be theoretically performed using our chip design. This expands the range of potential applications of this chip in extreme environments. Besides, in the manuscript's conclusion, we explore potential avenues for further optimizing the chip's performance.

Figure R7. Simulation of the impact of wavelength shift on the performance of DNN. **a** Phase modulation different at different working wavelengths. **b** The training set and test set accuracy of the DNNs by various wavelength. **c** Magnified intensity pattern at ‘7’ channel at 517 nm, 532 nm and 547 nm incident wavelength, scale bar: 200 μm . The white circle denotes the target detection area. The red dashed circle denotes actual detection area required for different working wavelengths.

Figure R8. Simulated Fashion-MNIST recognition performance of the bilayer DNN chip. A training set and a test set with 1000 images are used. **a-b** Normalized confusion matrixes of training set result and test set result, respectively.

References

- [1] Goi E, Schoenhardt S, Gu M. Direct retrieval of Zernike-based pupil functions using integrated diffractive deep neural networks. *Nature Communications* 13, 7531 (2022).
- [2] Veli M, et al. Terahertz pulse shaping using diffractive surfaces. *Nature Communications* 12, 37 (2021).
- [3] Qian C, et al. Performing optical logic operations by a diffractive neural network. *Light: Science & Applications* 9, 59 (2020).
- [4] Chen Y, et al. All-analog photoelectronic chip for high-speed vision tasks. *Nature*, 623, 48 (2023).

Revised content:

Lines 280-292, Page 13, Main Document

Figures 5d-f, Page 14, Main Document

Lines 248-253, Page 12, Main Document

Figure S11, Page 22, Supporting Information

Comment 6: Besides the major concerns, I also have some minor comments below:

- a. All camera image panels in the figures should include a color bar for better interpretability.
- b. The phase masks in the supplementary fig. S7 would benefit from color bars normalized by 2π .

Response: We appreciate your professional suggestions. We have now included color bars for all camera images and phase masks. We hope that the revised manuscript can be easier for readers to read and understand.

Revised content:

Figures 2b, Page 7, Main Document

Figures 3c, Page 10, Main Document

Figures 6b, Page 16, Main Document

Figure S2, Page 12, Supporting Information

Figure S8, Page 19, Supporting Information

Figure S10, Page 21, Supporting Information

Figure S13, Page 24, Supporting Information

Reviewer #2

The paper presents the development of a compact and robust two-layer integrated millimeter-scale diffraction neural network (DNN) chip for optical inference in extreme environments. The use of photons as information carriers in DNNs is a significant advance. It enables reasoning at ultra-low power and at the speed of light, a significant advance in artificial intelligence hardware. The design with two diffraction layers engraved on a quartz wafer improves the reliability of the device. The expected service life at room temperature is extremely long, suggesting its potential for long-term use. The chip's stability at high temperatures and 100% accuracy in recognizing two types of digits are commendable, indicating its suitability for operation in extreme conditions.

Response: Thank you for your time and attention to our manuscript. We greatly appreciate your comments and valuable suggestions on this work. We address the points you raised below and have revised the original manuscript seriously.

Comment 1: While the 82% recognition accuracy across ten digit types is impressive, it represents a somewhat narrow scope of testing. Real-world applications often require processing more complex and diverse data sets.

Response: Thank you for raising this concern. In the revised manuscript, utilizing the same parameters, we conducted simulations aimed at tackling Fashion-MNIST recognition. This dataset contains 10 categories of fashion products, presenting larger complexity compared with MNIST dataset. The DNN achieve approximately 92.2% and 80.1% accuracy in training and test, respectively. Figure R9 illustrates the confusion matrix for these results.

Meanwhile, we also tried a non-recognition AI task with potential practical applications: phase imaging. Phase is invisible, and extracting phase information from light has important applications in life sciences[1], wavefront shaping[2], and other fields. In our

simulation, the input digital image is phase information (Figure R10a). We train a bilayer DNN based on the same parameter as in the experiment (Figure R10b). From the result (Figure R10c), we can see that the DNN can directly convert the input phase to intensity information to realize phase imaging.

For other AI tasks, several pioneering studies have demonstrated the potential applications of DNNs in various fields [3-8]. Our DNN operates on the same physical diffraction principles as aforementioned works. Therefore, with appropriate design and optimization, we believe that our chip can replace these DNNs to accomplish the same tasks as well. Meanwhile, in the conclusion of the main document, we discuss future ways to enhance the intelligence of the DNN based on our method, which may further help the DNN to handle more complex tasks.

Figure R9. Simulated Fashion-MNIST recognition performance of the bilayer DNN chip. a-b Normalized confusion matrixes of training set result and test set result.

Figure R10. Simulated phase imaging performance of the bilayer DNN chip. a. Schematic of the simulation model. **b** Phase distribution of the DNN. **c** Training and test set simulation results for phase imaging. The inputs are phase-encoded handwritten digital images, and the outputs show light intensity distribution.

References

- [1] Park J, et al. Artificial intelligence-enabled quantitative phase imaging methods for life sciences. *Nature Methods*, **20**, 1645–1660 (2023).
- [2] Goi E, Schoenhardt S, Gu M. Direct retrieval of Zernike-based pupil functions using integrated diffractive deep neural networks. *Nature Communications* 13, 7531 (2022).
- [3] Fu T, et al. Photonic machine learning with on-chip diffractive optics. *Nature Communications* 14, 70 (2023).
- [4] He C, et al. Pluggable multitask diffractive neural networks based on cascaded metasurfaces. *Opto-Electronic Advances*, 7, 230005 (2024).
- [5] Zheng Z, et al. Dual adaptive training of photonic neural networks. *Nature Machine Intelligence*, **5**, 1119–1129 (2023).
- [6] C. Liu, et al, A programmable diffractive deep neural network based on a digital-coding metasurface array. *Nature Electronics* 5, 113 (2022).
- [7] Goi E, et al. Nanoprinted high-neuron-density optical linear perceptrons performing near-infrared inference on a CMOS chip. *Light: Science & Applications* 10, 40 (2021).
- [8] Y. Chen, et al, All-analog photoelectronic chip for high-speed vision tasks. *Nature*, 623, 48 (2023).

Revised content:

Lines 248-260, Page 12, Main Document

Figure S11, Page 22, Supporting Information

Figure S12, Page 23, Supporting Information

Lines 374-390, Pages 17-18, Main Document

Comment 2: The article does not sufficiently address the chip's performance with different types of data or its adaptability to other AI tasks beyond digit recognition. This raises questions about its versatility and applicability in various artificial intelligence applications.

Response: Thank you for your instructive comment. In our response to Comment 1, we have supplemented the simulation results of the DNN chip performing Fashion-MNIST recognition and phase imaging. These results indicate that the DNN chip method can be used for various AI tasks. Currently, in addition to images recognition, DNNs have been used for beam shaping[1], logical computing[2] and data downscaling[3]. The DNNs in these endeavors have the same physical diffraction principles as our work. The differences lie in the parameters, training models or datasets. Therefore, based on the design principles as these works, we think that our chip thus could theoretically perform these tasks as well.

References

[1] Veli M, et al. Terahertz pulse shaping using diffractive surfaces. *Nature Communications* 12, 37 (2021).

[2] Qian C, et al. Performing optical logic operations by a diffractive neural network. *Light: Science & Applications* 9, 59 (2020).

[3] Chen Y, et al. All-analog photoelectronic chip for high-speed vision tasks. *Nature*, 623, 48 (2023).

Revised content:

Lines 248-260, Page 12, Main Document

Figure S11, Page 22, Supporting Information

Figure S12, Page 23, Supporting Information

Lines 365-390, Pages 17-18, Main Document

Comment 3: The compactness of the device is an advantage, but the article does not address the challenges of scaling this technology to larger networks or integrating it with existing systems and technologies.

Response: Thank you for your professional comment. We wholeheartedly agree that implementing larger-scale networks and integrating with existing systems are important to further advance the practicality of this device. In the revised paper, we discuss feasible future technical routes based on our work.

For achieving larger networks, this involves larger numbers of neurons in each layer and more diffractive layers. The photolithography used in our work can currently achieve wafer-scale exposure, making it possible to increase the number of neurons. Then, to integrate more diffractive layers, we can further consider using existing bonding techniques. Bonding technology can bond multiple substrates together. For quartz substrates, laser bonding techniques can be a good solution [1,2]. It can fuse local regions of quartz, thereby enabling selective bonding of two quartz plates so that the structure of the diffractive layers is not affected in this process.

For system integration, there are pioneering endeavors to learn from. The DNNs chips can be integrated with camera chips [3,4]. In addition, researchers have recently reported the integration of DNN chip and electrical neural network chip, demonstrating an analog programmable optoelectronic chip [5]. Besides, we can also consider building a 3D integrated DNN system based on the vertical-cavity surface-emitting lasers (VCSELs) [6]. VCSELs are micron-sized on-chip light sources that emit light perpendicular to the substrate, so VCSEL can be used to generate optical images by constructing a two-dimensional addressable array. It is possible to integrate the DNNs structure directly on the surface of VCSELs by heterogeneous bonding [7]. Therefore, it is technically possible to realize a fully system integrated DNN chip.

References

[1] Huang H, Yang L-M, Liu J. Ultrashort pulsed fiber laser welding and sealing of transparent materials. *Appl Opt* 51, 2979-2986 (2012).

- [2] Zimmermann F, Richter S, Döring S, Tünnermann A, Nolte S. Ultrastable bonding of glass with femtosecond laser bursts. *Appl Opt* 52, 1149-1154 (2013).
- [3] Luo X, et al. Metasurface-enabled on-chip multiplexed diffractive neural networks in the visible. *Light: Science & Applications* 11, 158 (2022).
- [4] Goi E, et al. Nanoprinted high-neuron-density optical linear perceptrons performing near-infrared inference on a CMOS chip. *Light: Science & Applications* 10, 40 (2021).
- [5] Chen Y, et al. All-analog photoelectronic chip for high-speed vision tasks. *Nature*, 623, 48 (2023).
- [6] Gu M, Dong Y, Yu H, Luan H, Zhang Q. Perspective on 3D vertically-integrated photonic neural networks based on VCSEL arrays. *Nanophotonics* 12, 827-832 (2023).
- [7] Okumura K, Higurashi E, Suga T, Hagiwara K. Low-temperature GaAs/SiC wafer bonding with Au thin film for high-power semiconductor lasers. In: 2014 International Conference on Electronics Packaging (ICEP) (2014).

Revised content:

Lines 374-382, Page 17, Main Document

Lines 394-404, Page 18, Main Document

Comment 4: To summarize, while the development of this DNN chip represents a remarkable advance in artificial intelligence applications in extreme environments, there are several areas that require further study and clarification.

This article would be better suited for a more specialized journal.

Response: We greatly appreciate your valuable and constructive comments, which have greatly helped to improve the quality of our manuscript. We have seriously revised the manuscript according to these comments.

Based on the comments of all reviewers, we have made a comprehensive and detailed revision of the manuscript. Our revisions can be mainly summarized in the following aspects: (1) Based on the same parameters, we have added the simulation results of more types of tasks, including: Fashion MNIST dataset recognition and phase imaging. These results prove that our chip is a general integration strategy for DNNs that can be applied to a variety of AI tasks. (2) We conducted a comprehensive robustness analysis

of the chip. Including: the impact of double-sided lithography alignment error (Figure 5a and b, Main Document), the impact of chip thickness error (Figure 5b, Main Document), the impact of wavelength shift (Figure 5d-e, Main Document), the impact of input layer misalignment (Figure 5g, Main Document), and the impact of chip performance under unexpected mechanical damage (Figure S13, Supporting Information). (3) We analyzed the impact of various extreme environments on the performance of the chip (Supplementary Note 5, Supporting Information). (4) We conducted an in-depth discussion on the feasible routes for further system integration and performance optimization (Conclusion, Main Document).

To summarize the progress of our work, we have achieved vertical integration of the diffractive layers in the DNN on both sides of a single-crystal quartz substrate. Our approach provides a commercial and mature integration solution for 3D DNN chips, and demonstrates high robustness and consistency. We analyzed the lifetime of the DNN chip, which could become an important indicator for evaluating DNNs in practical applications. The ultra-long lifetime makes it suitable for performing various tasks in extreme environments with high stability.

Our progress in the integration, design, and robustness study of DNNs is focused on engineering. We sincerely hope that the revised manuscript can meet the high standards of *Communications Engineering*.

Reviewer #3

In this manuscript, the authors present a proposal for a millimeter-scale, compact, and robust dual-layer integrated deep neural network chip with a nearly infinite lifespan. They conducted practical experiments using quartz substrates and demonstrated excellent performance stability even at high temperatures. The innovation is intriguing and well-supported theoretically, and the overall paper is well articulated. The manuscript can be accepted with the following revisions:

Response: Thank you for your time and attention to our manuscript. We appreciate your positive comments which encourage us so much. Your detailed and professional critiques and advices are very helpful to us.

Comment 1: In Figs. 2 and 3, despite achieving a high level of training accuracy, there

appears to be some crosstalk in various receiver channels. Could the authors provide a brief analysis and propose corresponding improvement strategies?

Response: Thank you for your insightful comment. The cause of crosstalk is the binary phase modulation used, resulting in low diffraction efficiency. As a result, the zero-order diffraction represented by the digital image appears on the output layer. Minimizing crosstalk stands as a crucial factor in fortifying the consistency and robustness of DNN chips. We attempted to optimize the loss function to reduce the crosstalk (Figure S2, Supporting Information). However, as constraints increase, the accuracy of DNNs decrease as well. Therefore, in the experiment, we give priority to ensure the accuracy.

In the follow-up work, we will attempt several strategies to reduce the crosstalk. To start, employing a multi-layered DNN chip with multi-level phase modulation yields a reduction in the loss function value, and thus the crosstalk can also be reduced. Additionally, optimizing the DNN chip for smaller neuron feature sizes, namely the reduced pixel size, significantly amplifies the spatial bandwidth product and can minimize crosstalk. In addition, optimizing layer spacing also helps to improve the connectivity of network, thereby reducing the crosstalk.

Revised content:

Lines 164-172, Page 8, Main Document

Comment 2: The authors claim that "In addition, to reduce the noise in the output layer, the amplitude field in each region was optimized to a Gaussian distribution to increase the maximum light intensity." Could the authors further elaborate on the principle behind this approach?

Response: Thank you for your professional question. We found our expression is not rigorous and have revised it. The camera is composed of photodetectors, which have a certain sensitivity. Only when the photocurrent signal is higher than their detection threshold, we can observe the light signal on the camera's driver software. The area detecting a Gaussian distribution tends to exhibit a more concentrated intensity compared to typical uniform distribution area, thereby increasing the maximum light intensity density in the target region. As such, the effective signals can be easier to be captured by the camera, which enables cameras to operate with shorter exposure times

and/or under a lower-laser-power configuration. Thus, the noise in the recorded images can be reduced. Besides, reducing the laser power helps to reduce the energy consumption of the DNNs system.

Revised content:

Lines 139-145, Page 7, Main Document

Comment 3: Quartz substrates are employed for diffraction propagation in this manuscript. Could the authors briefly analyze how mechanical errors in the thickness of the quartz substrate affect the experimental results?

Response: Thank you for your instructive advice. The changes in the thickness of the quartz substrate will lead to changes in the layer spacing, which will have an impact on the performance of DNNs. We simulated the impact of the substrate thickness on the accuracy of DNNs. From Figure R11, it can be seen that the accuracy gradually decreases with the thickness deviation, but the rate of decrease is slow. When the thickness error is $\pm 50 \mu\text{m}$, the accuracy decreases by only about 2%. Therefore, we think that the effect of smaller thickness errors ($<10\%$) of the substrate on the performance of DNNs is negligible.

Figure R11. Simulation of the impact of thickness deviation of the quartz substrate on the performance of DNNs. a Schematic of the simulation model. b Accuracy of DNNs as a function of the thickness of the quartz substrate.

Revised content:

Lines 276-280, Page 13, Main Document

Figure 5c, Page 14, Main Document

Comment 4: The paper's title suggests permanence in extreme environments. In such scenarios, it is essential to analyze unexpected incidents beyond temperature indices, such as the impact of mechanical damage to the device. Please provide a brief robust analysis of these aspects.

Response: Thank you for your professional comment. For minor mechanical damage, like light wear, the surface of the sample may become rough. This effect is similar to high temperature annealing used for lifetime measurement in our experiment. After annealing, the surface roughness of the sample increases, and the effect is similar to mild mechanical wear. Experimental results show that this change has no impact on the accuracy of DNNs (Figure 6 in the Main Document). At the same time, we simulated the impact of severe mechanical damage on DNNs. A typical situation is that severe wear or sample fragmentation leads to the loss of part of neurons in the DNNs. As shown below, we modeled this kind of mechanical damage by erasing the phase information of a triangular region in the phase masks of DNNs. In Figure R12, the accuracy drop is depicted in relation to the increasing damaged area. Notably, the DNN chip can maintain its highest performance when the damage area is below 20%. This resilience can be attributed to the relatively small size of the designed input digits, which is half the size of the chip. Consequently, damage to the corners of the DNN chip has a minimal impact on performance. However, as the damaged area expands, the accuracy experiences a more rapid decline. The descent in accuracy accelerates proportionally with the growth of the damaged region.

Figure R12. Simulation of the mechanical damage impact of DNN chip. **a** Schematic of partial neuron loss in the sample caused by mechanical damage. **b** Accuracy of the DNNs changes with the damage area. The inset shows that we modeled two different shapes of damage.

Revised content:

Lines 346-350, Page 16, Main Document

Figure S14, Page 25, Supporting Information

Comment 5: I suggest that the authors cite some representative papers on metamaterials designed by machine learning and deep learning, including [Nat Commun 13, 2694 (2022)], [Light Sci Appl 12, 82 (2023)], [Prog Electromagn Res 175, 139-147, (2022)], [Prog Electromagn Res 175, 81-89, (2022)]. This will help contextualize their work within the existing literature and provide additional support for the novelty and significance of their contributions.

Response: Thank you for your valuable comment. We wholeheartedly agree that these advances hold significant importance. Consequently, we have incorporated them into the manuscript. Meanwhile, more important related works are also cited.

Revised content:

References, Pages 20-21

Ref. [3]: Jia Y, Qian C, Fan Z, Cai T, Li E-P, Chen H. A knowledge-inherited learning for intelligent metasurface design and assembly. *Light: Science & Applications* **12**, 82 (2023).

Ref. [4]: Shou Y, Feng Y, Zhang Y, Chen H, Qian H. Deep learning approach based optical edge detection using ENZ layers. *Progress In Electromagnetics Research* **175**, 81-89 (2022).

Ref. [9]: Tan Q, Qian C, Cai T, Zheng B, Chen H. Solving multivariable equations with tandem metamaterial kernels. *Progress In Electromagnetics Research* **175**, 139-147 (2022).

Ref. [19]: Qian C, *et al.* Dynamic recognition and mirage using neuro-metamaterials.

Nature Communications **13**, 2694 (2022).

Other newly cited references include Ref. [10-13] and Ref. [22].

Finally, we thank you once again for the review of our manuscript and the constructive comments.

Yours sincerely,

Min Gu on behalf of all contributing authors

REVIEWERS' COMMENTS:

Reviewer #1 (Remarks to the Author):

All my comments have been addressed. I do not have any more comments to add. I support its publication.

Reviewer #3 (Remarks to the Author):

Having thoroughly examined the author's responses to all comments and suggestions, it is apparent that the author has approached each of the reviewers' concerns with diligence. The revisions made by the author reflect a commendable effort in enhancing the overall quality of the manuscript. I recommend the publication of this paper.

Manuscript ID: COMMSENG-23-0475B

Title: Compact eternal diffractive neural network chip for extreme environments

Reviewer #1

All my comments have been addressed. I do not have any more comments to add. I support its publication.

Response: Thank you for your time and attention to our manuscript. Your comments have helped us to improve the quality of the manuscript. We greatly appreciate your recommendation.

Reviewer #3

Having thoroughly examined the author's responses to all comments and suggestions, it is apparent that the author has approached each of the reviewers' concerns with diligence. The revisions made by the author reflect a commendable effort in enhancing the overall quality of the manuscript. I recommend the publication of this paper.

Response: Thank you for your review of our manuscript. Your suggestions helped us to improve the quality of the manuscript. We greatly appreciate your recommendation.

Finally, we thank you once again for the review of our manuscript and the constructive comments.

Yours sincerely,

Min Gu on behalf of all contributing authors